# The Development and Characterisation of A Porcine Large Intestinal Biological Scaffold by Perfusion Decellularisation

**DOI:** 10.3390/cells14110817

**Published:** 2025-05-31

**Authors:** Murali Somasundaram, Karin V. Greco, Gauraang Bhatnagar, Simon Gabe, Paul Sibbons, Peter Friend, Tahera Ansari

**Affiliations:** 1Research and Development Department, Northwick Park Institute for Medical Research—The Griffin Institute, London HA1 3UJ, UK; 2Nuffield Department of Surgical Sciences, Oxford Transplant Centre, University of Oxford, Oxford OX1 4BH, UK; 3Division of Surgery and Interventional Science, Royal Free Hospital Campus, University College London (UCL), London NW3 2QG, UK; 4Department of Radiology, St Mark’s Hospital and Academic Institute, London NW10 7NS, UK; 5Lennard-Jones Intestinal Failure Unit, St Mark’s Hospital and Academic Institute, London NW10 7NS, UK

**Keywords:** tissue-engineered large intestine (TELI), porcine model, in vivo, full-thickness large intestinal scaffolds, perfusable native vasculature, xenogeneic source

## Abstract

The rising incidence of colorectal cancer and ulcerative colitis underscores an urgent need for regenerative solutions to address functional deficits after colectomy. However, the creation of clinically applicable large intestine scaffolds remains underdeveloped. Here, we report the successful generation and thorough characterisation of transplantable-sized porcine large intestinal scaffolds via perfusion decellularisation. This method effectively preserved extracellular matrix (ECM) structural and biochemical integrity while minimising immunogenicity through cellular component removal. Crucially, native vasculature remained intact, confirmed by histology, DNA quantification, and high-resolution CT angiography. Despite efficient decellularisation, challenges including residual nucleic acids, ECM heterogeneity, and partial microvascular occlusion were noted, echoing ongoing limitations in engineered, perfusable, full-thickness scaffolds. In vivo implantation demonstrated favourable biocompatibility and host integration; however, thrombosis occurred due to the lack of pre-seeded cells, emphasising the necessity of recellularisation for functional perfusion prior to implantation. This study addresses significant field limitations, presenting the first reproducible approach for structurally intact, perfusable, full-thickness large intestinal scaffolds of transplantable dimensions. Our innovations offer a strong foundation for future integration of patient-derived cells, stem cells, and organoids, progressing toward clinically viable, scalable, tissue-engineered large intestine constructs, from xenogeneic sources, relevant for regenerative medicine, disease modelling, and pharmacological screening.

## 1. Introduction

Over recent decades, regenerative medicine has emerged as a multidisciplinary frontier, integrating stem cell biology, biomaterials science, and tissue engineering to address the unmet clinical needs posed by organ failure and complex tissue loss [1,2,3,4,5]. Among its applications, intestinal tissue engineering (ITE) presents unique challenges due to the structural complexity, layered histology, and critical absorptive and barrier functions of the gut [5]. Colorectal cancer, ulcerative colitis, and short bowel syndrome (SBS) frequently necessitate partial or total colectomy, resulting in major functional impairments including diarrhoea, dehydration, and long-term dependence on parenteral support or permanent stomas [6].

ITE offers a promising solution to these challenges. Nevertheless, despite a long-standing research interest, the most significant progress in ITE has disproportionately concentrated on the small intestine [1,7,8,9,10,11]. The most efforts have been restricted to epithelial monolayers, mucosal patches, microscale scaffolds, or scaffolding material mimicking a 3D bowel structure in the few large intestine studies [12,13,14]. Yet, the clinical potential of a full-thickness, perfusable tissue-engineered large intestine (TELI) is substantial—it could restore water absorption, faecal consistency, and host–microbiome homeostasis, while also serving as a platform for disease modelling and drug testing [15,16,17,18]. 

The saga of finding efficacious therapies for the intestinal replacement has propelled relevant studies in the tissue engineering/regenerative medicine field. Kitano and colleagues (2017) demonstrated that human iPSC-derived intestinal epithelium and endothelial cells could repopulate decellularised intestinal matrices to form perfusable, absorptive grafts [11]. Meran and collaborators (2023) extended this work to the human scale, engineering mucosal grafts with patient-derived organoids and fibroblasts seeded onto decellularised human intestine [10]. Tullie and colleagues (2022), additionally, underscored the necessity of recapitulating not just epithelium, but also vasculature, mesenchyme, and innervation to produce clinically relevant intestinal grafts [1].

Our study responds to this challenge with a novel focus on the large intestine, using porcine colon—a rarely addressed but clinically significant target in ITE. We developed a transplantable-sized biological scaffold via vascular perfusion decellularisation, preserving both the native extracellular matrix (ECM) and the attached vasculature, including mesenteric arterial and venous structures. This anatomical continuity is critical: Blood vessels, together with organ parenchyma, form the structural and functional backbone of most organs and are essential for translational use in perfusion, recellularisation [19,20,21], and transplantation workflows [22].

Perfusion decellularisation was adapted from protocols successfully applied to small intestine, heart, liver, muscle, lung, and kidney [15,16,17,18,19,20] and has been rigorously optimised for full-thickness vascularised large intestinal scaffolds of transplantable dimensions. 

Scaffold integrity was assessed through histological staining (H&E, Masson’s Trichrome, Alcian Blue, Picro-sirius Red), DNA quantification, and transmission electron microscopy. Vascular patency was evaluated using dye perfusion and computed tomography angiography (CTA).

By leveraging xenogeneic tissues, we established a scalable, ethically sustainable platform for scaffold production. This strategy not only bypasses the critical shortage of human donor organs—preserving them for life-saving transplantation—but also aligns with state-of-the-art regenerative medicine practices. Notably, ECM components are highly conserved across species, making xenogeneic scaffolds biologically compatible and translationally relevant [23], while providing a robust foundation for an “off-the-shelf” clinical-scale scaffold production without depleting finite human tissue resources.

## 2. Materials and Methods

### 2.1. Intestinal Retrieval and Bench Preparation

Colon was retrieved after cardiac death in crossbred white Landrace pigs. All procedures were conducted in accordance with Home Office Animals (Scientific Procedures) ACT approval as per UK Guidelines PPL 80/1752 and PPL 80/2032 and following ethical permission from the institutes’ ethical review body. Following termination, abdominal access was gained by laparotomy, and colon distal to the spiral portion was identified. The inferior mesenteric artery and vein were dissected, and a 20 cm specimen of colon was removed with supplying vessel and aortic trunk en-bloc. The entire specimen was then bench prepared by cannulating the aorta and perfusing with 3 L of heparinised N. Saline (25,000 U/L) by gravity. During perfusion, redundant arterial vessels were ligated, permitting isolated perfusion of the intestine and creating an arterio-venous flow circuit. Venous outflow was then cannulated, and the entire specimen washed in warm water to remove faecal residues. The proximal colon was cannulated with 7 cm long silicon tube (3.6 mm diameter). A simple intra-arterial injection of 1:1 diluted iodine was used to identify any final leaks in arterial circuit before beginning decellularisation. 

### 2.2. Perfusion Decellularisation

Decellularisation was performed based on a previous study from our team [24]. Briefly, it was achieved by peristaltic pump perfusion (Watson Marlow 323 U/D) at a rate of 30 rpm. A double perfusion circuit was established, permitting the inflow of reagents into the colonic lumen via the silicon tubing and the arterial supply via the aortic cannula. Perfusion fluid drained at the superior aspect of the specimen via the inferior mesenteric vein. During the process, the entire specimen was immersed in decellularisation reagents that served as a reservoir. An illustration of the perfusion circuit can be seen in Figure 1.

Before beginning decellularisation, the specimen was perfused with de-ionised water to remove tissue and cellular debris. A series of protocols were trialled in a number of retrieved colon specimens from pigs (n = 20) in order to establish a method of decellularising the intestine whilst preserving the architecture of the ECM as intactly as possible. Reagents that proved effective included Trypsin, EDTA, DNase, and sodium dodecylsulfate (SDS) (Sigma Aldrich, Poole, UK). The precise concentrations, duration, and use of these reagents is listed in Table 1. Note that the use of reagents was grouped into a series of steps to form cycles which were repeated. The use of intermittent washes with distilled water (both at room temperature) enabled the removal of cellular debris. The efficacy of different decellularisation protocols was assessed using Haematoxylin and Eosin (H&E) staining. A number of specimens of retrieved colon (n = 10) were allocated for the optimisation of decellularisation protocol and divided into groups of fresh (n = 4), frozen for 3 months (n = 3), and frozen for 6 months (n = 3). Samples were frozen at −20 °C without PBS or any additional solution in sterile cryo-containers prior to the initiation of the perfusion decellularisation protocol. This freezing step aimed to promote ice crystal formation, contributing to the subsequent decellularisation process. The step also assessed the possibility of an “off-the-shelf” approach to decellularisation. The frozen samples were then thawed at appropriate intervals (3 and 6 months as above) for the comparison of the effect of freezing on the perfusion decellularisation process detailed above. Following decellularisation, all specimens were preserved in phosphate buffered saline (PBS) and stored at 4 °C until further use (within 48 h). 

### 2.3. Histological Analysis

The specimens were removed from cold storage and the vascular pedicle dissected free. The 20 cm colonic segment was divided into 5 × 4 cm segments from proximal to distal labelled sequentially and numerically. An amount of 1 cm^2^ sheets of full-thickness intestine/scaffold from each segment were excised for testing. Samples from each segment were excised and fixed for 24 h in 10% NBF solution at room temperature. Each tissue sample was processed for paraffin embedding using a standardised histological protocol. Briefly, the samples were fixed in 10% neutral buffered formalin for 24 h at room temperature, followed by dehydration in an ascending ethanol series: 70% ethanol (1 h), 90% ethanol (1 h), 100% ethanol (2 changes, 1 h each). This was followed by clearing in xylene (2 changes) and embedding in paraffin wax at 60 °C under vacuum conditions to ensure complete infiltration. The embedded tissues were allowed to cool at room temperature before sectioning. The paraffin blocks were sectioned into 5 µm slices using a rotary microtome. The sections were mounted onto glass slides, dried at 37 °C overnight, and subsequently stained with Haematoxylin and Eosin (H&E) to evaluate the presence of cells. Standard H&E staining was performed to evaluate tissue morphology and cellular presence. Briefly, the paraffin was removed with xylene (2 × 5 min), and the sections were rehydrated through a descending alcohol series (100%, 95%, 70%, distilled water). The sections were stained with Mayer’s haematoxylin for 5 min, rinsed in running tap water, differentiated in 1% acid alcohol, and counterstained with eosin for 2 min. The slides were dehydrated, cleared in xylene, and coverslipped with mounting medium.

Picro-sirius red with Millers elastin (PME) and Masson’s Trichrome were used to assess the components of the ECM. Masson’s Trichrome staining was performed following deparaffinisation and rehydration; the sections were stained with Bouin’s solution at 56 °C for 1 h to enhance staining quality. After cooling and rinsing in running tap water, the slides were stained with Weigert’s iron haematoxylin (10 min), rinsed, and then stained sequentially with Biebrich scarlet-acid fuchsin solution for 10 min. This was followed by phosphomolybdic-phosphotungstic acid treatment for 10 min, then aniline blue for 5 min. The slides were rinsed in distilled water, differentiated in 1% acetic acid, dehydrated, cleared, and coverslipped. The PME-stained samples were also observed under polarised light for a detailed analysis of the structural integrity of collagen fibres. Alcian Blue was used to assess the preservation of the glycosaminoglycans (GAGs) in the tissue following decellularisation when compared to the control tissue. The samples were analysed by using the ZEISS Axio Imager 2 microscope (Oberkochen, Germany) at 20 and 40× magnification.

### 2.4. Immunohistochemical Analysis

The sections were prepared as described above but on 3 Aminopropyltriethoxysilane (APTS) coated slides. The slides were stained for Major Histocompatibility Complex II (MHC II). In brief, antigen retrieval was necessary using Citrate Buffer (10 mM) at 95 °C for 15 min, and staining was performed using ImmPRESS™ kit (Vector Labs, Newark, CA, USA). Following endogenous and non-specific blocks, primary antibody incubation (MHC II [mouse anti-pig, H42A, VMRD, UK] 1:100 dilution) took place at room temperature for 2 h before the addition of ImmPRESS kit™ according to the manufacturer’s guidelines. The slides were counterstained with Harris’ haematoxylin.

### 2.5. Molecular Analysis

#### 2.5.1. DNA Quantification

The assessment of residual DNA within specimens was quantified using commercial assay kits (GenElute™ Mammalian Genomic DNA Miniprep Kit, Sigma Aldrich, Poole, UK). Following the decellularisation and cut-up described above, the specimens were snap frozen in liquid nitrogen and preserved at −80 °C for future analysis. The manufacturer’s protocols were followed. In brief, 25 mg samples underwent mechanical disruption, dissolution in lysis buffer, and extraction in buffer before precipitation in ethanol. Total DNA quantification was performed using a UV-Vis spectrophotometer (NanoDrop ND1000, Thermo Scientific, Wilmington, DE, USA). Absorbance was measured at 260 nm, which corresponds to the peak absorbance wavelength for nucleic acids, and the absolute amount of DNA per milligram of tissue (ng/mg) was calculated.

#### 2.5.2. Glycosaminoglycan (GAG) Quantification

To quantify GAG content in both the control and decellularised intestine, the Blyscan GAG assay kit (Biocolor, Carrickfergus, Northern Ireland) was used. In brief, 50 mg of minced wet tissue was placed in a micro-centrifuge tube and incubated with 1 mL of papain digestion buffer at 65 °C for 18 h. Aliquots of each sample were mixed with 1,9-dimethyl-methylene blue (DMMB) dye and reagents from the GAG assay kit, and 200 µL of each sample was added in triplicate in a 96-well plate. The absorbance was measured using a plate reader (Versamax, Molecular Devices LLC, San Jose, CA, USA) at 656 nm, and the absolute GAG content was calculated per milligram of tissue.

#### 2.5.3. Transmission Electron Microscopy (TEM)

The rehydrated scaffold and control samples were fixed in 3% glutaraldehyde in 0.1 M sodium phosphate buffer (pH 7.4) at room temperature. Subsequent fixation for 1 h was performed using 1% osmium tetroxide in 0.1 M sodium phosphate buffer (pH 7.4). The samples were then washed with distilled water and block stained in 2% Uranyl acetate (Agar Scientific, Stansted, UK) for 2–4 h. Following this, the samples were then washed with distilled water and gradually dehydrated using an acetone gradient and gradually infiltrated with araldite resin (Agar Scientific, Stansted, UK). After infiltrating for 8 h with two changes of araldite, the samples were embedded in araldite, cured for 18 h at 65 °C, and sectioned. The ultrathin sections (100 nm thick; using a Reichert–Jung Ultracut E microtome, Vancouver, Canada) were collected on 200 mesh copper grids (Agar Scientific, Stansted, UK), stained with Reynold’s Lead citrate, and carbon coated. The specimens were viewed using a Jeol JEM-1200 EX electron microscope (Tokyo, Japan).

### 2.6. In Vivo Biocompatibility

The scaffold segments were sterilised in 0.1% peracetic acid by immersion. Subcutaneous implantation of decellularised and control intestine was performed in Lewis rats (n = 10). All the animal procedures complied with institutional ethical use protocols (as stated in the NIH Guide for Care and Use of Laboratory Animals). On the day of surgery, rats were anaesthetised using sodium pentobarbital administered intraperitoneally at a dose of 50 mg/kg body weight and placed in the supine position. A midline abdominal skin incision was made, and 2 scaffold samples and 2 controls were implanted subcutaneously in each animal. The location of the implants was randomly allocated, and implants were secured to the abdominal wall muscle by sutures (for later identification). The skin was closed, and the rats recovered. Half of the rats were terminated at 2 weeks, and the remainder at 4 weeks by lethal injection. On termination, a lethal dose of sodium pentobarbital (≥150 mg/kg, intraperitoneally) was administered following experimental procedures, in accordance with UK Home Office standards. The incision was reopened, the implants identified, and photographic images taken. The implants were removed with the underlying muscle layer attached to enable orientation, were fixed in 10% NBF for 48 h, and were processed for histological analysis.

### 2.7. Perfusion Studies

#### 2.7.1. Intraluminal Patency

Patency of scaffold lumen was assessed by a simple intra-luminal injection of 10% iodine solution with occlusion of the distal end by ligature.

#### 2.7.2. Vascular Dye Injection Studies

The initial assessment of vascular patency was performed by an intra-arterial injection of Trypan blue dye diluted in water (1:10) or povo-iodine solution (1:10). Standard photographic images were obtained once the dye had perfused throughout the specimen.

#### 2.7.3. Vascular Transplantation

Following decellularisation, a specimen was stored in PBS while a recipient pig was prepared for transplantation. Intra-abdominal access was gained by laparotomy, and the right kidney ligated at the renal pelvis. The supplying renal artery and vein were dissected to provide appropriate-sized vessels for anastomoses. Continuous 6.0 prolene venous and arterial anastomoses were performed using standard vascular surgery techniques, and the clamps removed. The specimens were imaged on reperfusion and reviewed over a 60 min period. Following termination, the specimens were removed and processed by routine histological techniques as above.

#### 2.7.4. Computerised Tomographic (CT) Angiography

The specimens were placed in a plastic receptacle, the colonic lumen filled with water (approx. 500 mL), and the cannulas clamped at proximal and distal ends to distend the specimen and improve image quality. An amount of 10 mL of Visipaque 270 contrast diluted 1:10 in distilled water volume was then injected intra-arterially, and images taken using a Philips INGENUITY (128-slice) scanner at 80 kV and 20 mA (pitch of 1), equating to 20 mAs with a FoV (Field of View) 441 mm.

## 3. Results

### 3.1. Specimen Retrieval and Assessment of Decellularisation

Porcine colon specimens were successfully removed with the vascular pedicle in continuity. Bench preparation ensured arteriovenous vascular flow, and decellularisation was performed as exemplified in Figure 1, resulting in a scaffold with a translucent gelatinous structure illustrated in Figure 2a. Samples were obtained from ten individual White Landrace pigs. Multiple samples were taken from each animal and randomised across all experimental groups to achieve the 20 replicates reported. These samples from decellularised scaffolds were subdivided across different analyses for assessment of the presence of cells, and tests to assess ECM integrity, based on the protocol requirements of each procedure. The protocol was reproducible across all animals. The intestinal scaffold tissue was maintained in continuity without breaches as demonstrated by the filling of lumen with 10% iodine contrast (Figure 2b).

Assessment of residual cellular material by H&E staining demonstrated an absence of cells following 5 cycles of decellularisation, which was reproducible following a series of experiments (n = 10). The absence of cells was evident in the decellularised samples (Figure 2d) when comparing the decellularised tissue to the controls (Figure 2c). Upon histological analysis, no difference in the presence of cells was noted when comparing proximal to distal segments of colon, indicating the consistent removal of cells throughout the specimens.

Following 5 cycles of decellularisation, the scaffolds were stored at −80 °C, and DNA extraction and quantification performed as described. DNA was extracted from 5 separate scaffolds (labelled DC 1-5) and compared with the control colon tissue. Following decellularisation, a significant reduction (* *p* = 0.045) in DNA was noted with the mean value measuring 202 ± 79.6 ng/mg in contrast to the control colon, which contained 893.9 ng/mg (Figure 2e).

To assess the potential impact of freezing prior to decellularisation (DC), the scaffolds were either processed immediately or stored at −20 °C for 3 or 6 months prior to the DC process. All DC groups, including the fresh and frozen samples, showed a statistically significant reduction in DNA content compared to the control (* *p* = 0.035). However, no significant differences were observed between the fresh and frozen DC groups, indicating that freezing did not alter the extent of DNA removal in these experimental settings (Figure 2f).

### 3.2. Assessment of ECM Architecture 

Different aspects of the ECM were analysed as described. Collagen was assessed initially by PME staining, viewed under both standard and polarised light (Figure 3a–e). The PME-stained sections demonstrated the retention of collagen fibres, which showed to be similar to the control. The ultrastructure of the scaffolds was assessed by TEM, with the control images clearly demonstrating cells, nuclei, and the ECM at magnification of 6900× (Figure 3c). TEM images of the scaffolds were characterised by an absence of cells with islands of structurally arranged filamentous tissue consistent with collagen demonstrating the preservation of ECM architecture following decellularisation (×27,600) (Figure 3f). Masson’s Trichrome staining provided alternative imaging of ECM components, showing positive staining for elastin (Figure 3g,h), an indicative of the preservation of these proteins as the main structural part of the ECM in intestinal tissues.

Alcian Blue staining identified preserved GAGs within the scaffolds in comparison to the controls (Figure 3i,j). The quantification of GAGs showed that decellularisation led to an approximate 75% reduction in GAGs in the scaffolds and the effect of freezing led to a further reduction (Figure 3k). While we recognise this method may impact GAG content due to molecular leakage or entrapment in damaged matrix during freezing cycles, it was standardised across all conditions and is acknowledged as a technical variable. It is of note that the quantity of GAGs (µg/mg) was low, even in the control tissue (0.12 µg/mg), reinforcing the effect of freezing on matrix preservation, and this data was validated by the assessment of GAG quantification in other control tissues such as dermis and cartilage. While Figure 3k visually suggests some variation in GAG levels across different freezing durations, statistical analysis revealed no significant differences between the 3- and 6-month frozen groups.

### 3.3. Scaffold Immunogenicity and Biocompatibility

The absence of cells in the scaffold material (as per H&E staining, Figure 2c) negated any cell membrane positivity to MHC II-immunohistochemical staining in the decellularised tissue (Figure 2h), whereas the control tissue demonstrated the classical MHC II expression (Figure 2g). The histological results were corroborated by an in vivo biocompatibility test in rats, with a 100% survival rate after 4 weeks. There appeared to be an obvious difference in macroscopic appearance, indicating greater inflammatory and immune reaction in areas of the control tissue implantation when compared to the scaffolds (Figure 4a). After 4 weeks, the scaffolds and control tissue were noted to have macroscopically completely degraded/remodelled (Figure 4d). The histological analysis demonstrated a clear difference in the scaffold’s appearance when compared to that of the controls after 4 weeks. Specifically, we highlight that cellular infiltration appeared notably higher in the control (non-decellularised; Figure 4b,c,e) implants compared to the decellularised scaffolds, with minimal cellular infiltrate (Figure 4f). This observation is consistent with an immune response typically triggered by the presence of native cellular components, as in the non-decellularised tissues.

### 3.4. Vascular Transplantation and CT Angiographic Evaluation

Vascular transplantation of the scaffolds (n = 2) was conducted via end-to-end anastomosis with the renal artery, demonstrating immediate perfusion (Figure 5a). Perfusion into small vessels was clearly observable (Figure 5b); however, effective scaffold perfusion was transient due to thrombosis occurring approximately 17 min post-anastomosis, resulting in thrombotic occlusion of venous drainage (Figure 5c, iii). CT angiography subsequently confirmed full perfusion within the small vessels of the scaffolds (Figure 5d), while contrast-enhanced CT images identified small vessel perfusion accompanied by minor leakage points (Figure 5e).

### 3.5. Perfusion Studies and Vascular Integrity

Perfusion was initially assessed by the intra-arterial injection of aqueous trypan blue, revealing vessel perfusion along with leaks from smaller vessels. Detailed examination through CT angiography verified contrast-effective flow through arterial and venous systems. However, despite successful initial reperfusion and the maintained patency of larger vessels, small vessel leaks manifested as extravasation (“blushing lesions”) (Figure 5d,e). Physiological directional flow persisted despite these leaks. Additionally, perfusion extended into small vessel networks within the intestinal wall (Figure 5b), yet consistent with implantation outcomes in “bare scaffolds”, thrombotic events compromised venous drainage.

## 4. Discussion

Despite longstanding interest in intestinal tissue engineering (ITE), most breakthroughs to date have unequally centred on the small intestine, with large intestinal constructs receiving comparatively little attention [1,7,8,9,10,11]. Prior efforts have often been limited to epithelial monolayers, mucosal substitutes, or microscale scaffold systems, with only a few studies extending these strategies to the complex architecture of the colon studies [12,13,14].

The reconstruction of complex, vascularised organs remains one of the most formidable challenges in regenerative medicine. While advances in tissue engineering have yielded promising results for simpler systems, the fabrication of full-thickness, perfusable constructs that recapitulate the intricate structure and function of the large intestine remains elusive [7,8].

Such constructs could restore the absorptive and barrier functions of the colon, re-establish host–microbiome balance, and serve as physiologically relevant models for disease investigation and drug testing [16,17].

Our study provides a significant step toward bridging this gap by demonstrating the reproducible development of transplantable-sized porcine large intestinal scaffolds that maintain native vasculature, the integrity of the most relevant parts of the extracellular matrix (ECM), and offer scalability from xenogeneic sources. These attributes are essential to enable a meaningful progression toward clinical-grade tissue-engineered large intestine (TELI) constructs.

By targeting the porcine colon distal to the spiral segment, we achieved anatomically consistent retrieval of vascularised tissue using standardised simple surgical techniques. This region enabled facile dissection and perfusion access, critical for consistent decellularisation outcomes. Importantly, we established that such tissues can be harvested post-cardiac death and stored frozen for extended durations without compromising ECM architecture. This widens the donor window, including the potential use of cadaveric tissue and circumventing constraints imposed by cold ischaemic time, thereby expanding potential donor pools and supporting the feasibility of scalable, off-the-shelf scaffold manufacturing [25,26,27].

The decellularisation strategy adapted from previous protocols [15,24] successfully removed cellular elements while preserving key ECM proteins such as collagen and elastin, as confirmed by histological staining and transmission electron microscopy. Such results have been widely reported in the literature for a number of tissues including heart, lung, liver, kidney, and trachea [15,16,17,18,28,29].

Nevertheless, DNA analysis revealed residual nucleic acid levels averaging 202 ± 79.6 ng/mg, exceeding the often-cited 50 ng/mg threshold [30,31,32]. This benchmark, though widely used, is not validated for intestinal tissue and may be overly conservative given the ECM complexity of the colon. Indeed, several studies suggest that antigenicity is governed not solely by DNA content, but also by epitope exposure and cell–matrix interaction dynamics [33,34,35].

Notably, glycosaminoglycan (GAG) content decreased post-decellularisation, which was pared with the control samples. This is consistent with similar tissue engineering studies demonstrating low quantities of GAGs both within decellularised and control tissue as a percentage of total weight [36]. Although apparent fluctuations in GAG levels were observed between samples frozen for 3 and 6 months, no statistically significant differences were identified. This suggests that while GAGs are crucial ECM constituents for biomechanical and signalling roles, their preservation is subject to biological variability and technical factors, including enzymatic degradation, leaching during decellularisation, microstructural alterations in the collagen network, potentially trapping or obscuring GAG molecules, and freeze–thaw instability. Moreover, the inherently low baseline GAG content in porcine colon aligns with prior reports, reinforcing the notion that ECM analysis must be tissue-specific and multidimensional [36].

Biocompatibility studies confirmed the scaffold’s safety profile. Subcutaneous implantation in rats elicited minimal inflammatory responses and no mortality. However, resorption of both the scaffolds and control tissue by week four highlighted the necessity of scaffold modification through crosslinking or cellular priming to support long-term integration and regeneration [37,38,39].

Vascular perfusion was preserved in major vessels but remained suboptimal within the smaller microvascular compartments. This limitation resulted in early thrombosis during in vivo transplantation, which curtailed the number of animals available for follow-up analysis and, therefore, should be considered as a proof-of-concept. Nonetheless, such outcomes align with observations in other decellularised scaffolds lacking endothelial coverage [24,39]. The application of computed tomography angiography (CTA) in this study, for the first time in a TELI model, provided non-invasive, high-resolution visualisation of vascular integrity and perfusion pathways. This innovation offers a translationally relevant platform for scaffold quality control prior to implantation.

Our data suggest that decellularised large intestinal scaffolds must undergo endothelialisation prior to clinical use to mitigate pro-thrombotic risk and improve microvascular patency. Future directions therefore include recellularisation using patient-derived organoids or induced pluripotent stem cells within bioreactor environments optimised for perfusion and tissue maturation [21,22,23,40,41,42].

By integrating vascular preservation, multimodal scaffold characterisation, and in vivo perfusion validation, this study lays a foundation for advanced recellularisation strategies using patient-derived cells, stem cell technologies, and organoid systems within a bespoke bioreactor environment. These innovations advance the field toward clinically viable, “off-the-shelf” tissue-engineered large intestine constructs from xenogeneic source.

## 5. Conclusions

This study demonstrates a reproducible and scalable approach for the generation of large intestinal scaffolds using perfusion decellularisation of porcine colon. Key achievements include the preservation of full-thickness ECM integrity, retention of mesenteric vasculature, and structural validation through histology, DNA quantification, and high-resolution imaging techniques, notably CT angiography. These attributes are crucial for supporting subsequent recellularisation and functional regeneration.

Despite promising outcomes, challenges persist. Residual DNA content, ECM heterogeneity, and early thrombosis post-implantation signal the need for further optimisation in decellularisation precision, endothelialisation, and scaffold conditioning. Nevertheless, in vivo studies confirm the scaffold’s favourable biocompatibility, offering a viable substrate for future bioengineering applications.

Employing xenogeneic tissue sources, this work established a reproducible, sustainable, and ethically responsible approach to scaffold generation, mitigating the reliance on human donor organs. As such, this study forms a critical foundation for the translation of large intestinal tissue engineering into therapeutic practice, contributing meaningfully to advances in regenerative medicine, disease modelling, and pharmacological testing.

## Figures and Tables

**Figure 1 cells-14-00817-f001:**
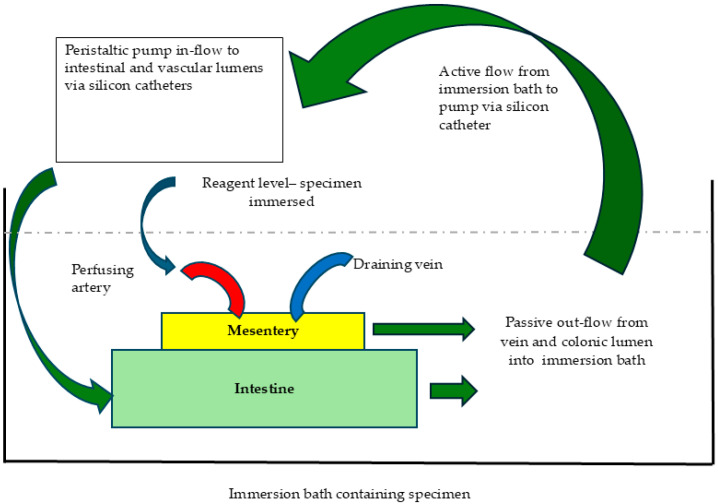
Schematic Representation of Perfusion Circuit. Illustration demonstrating the dual perfusion circuit supplying both intestinal and vascular lumens via silicon catheters. Venous and intestinal drainage catheters direct outflow into an immersion bath, from which solutions are recirculated.

**Figure 2 cells-14-00817-f002:**
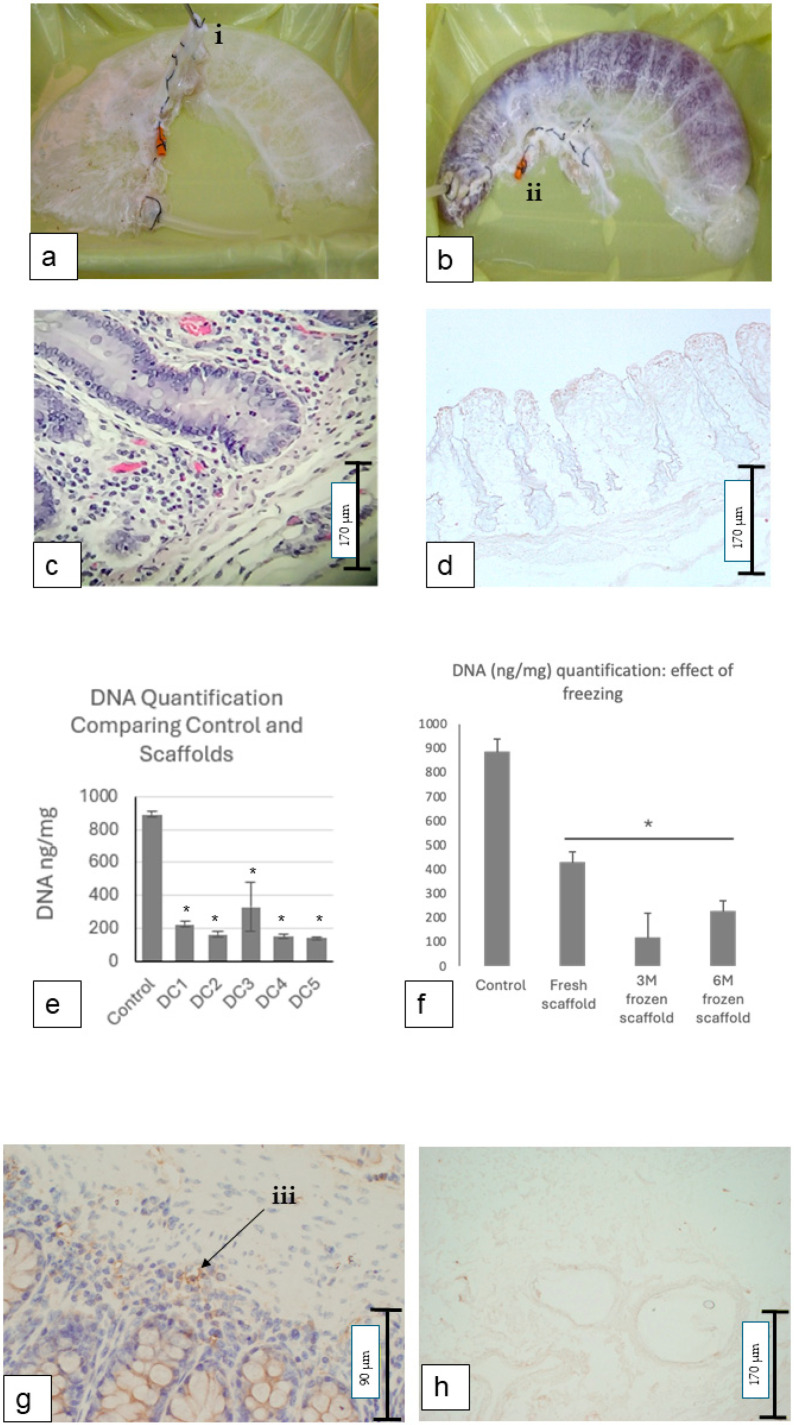
Scaffold Characterisation Post-decellularisation. (**a**) Macroscopic view of scaffold post-decellularisation, highlighting vascular aortic inflow (lifted by forceps, ‘i’); (**b**) scaffold lumen filled with contrast, confirming absence of leaks, with arterial cannula placement (‘ii’); (**c**) control intestinal tissue stained, showing clear mucosal and muscular layers; (**d**) decellularised colon with absence of cellular material; (**e**) quantitative DNA comparison between decellularised scaffolds (n = 10) and control tissues, showing consistent reduction in DNA throughout decellularised (DC) samples; (**f**) impact of freezing at −20 °C on decellularisation, depicting DNA reduction with some variability significantly diminished when compared to controls, though not differing among decellularised scaffolds, whether fresh or frozen for 3 or 6 months; (**g**) MHC II-positive staining (‘iii’) in control colon; and (**h**) absence of MHC II staining in decellularised scaffold, indicating removal of cellular components. * statistically significant reduction compared to the control (*p* < 0.05).

**Figure 3 cells-14-00817-f003:**
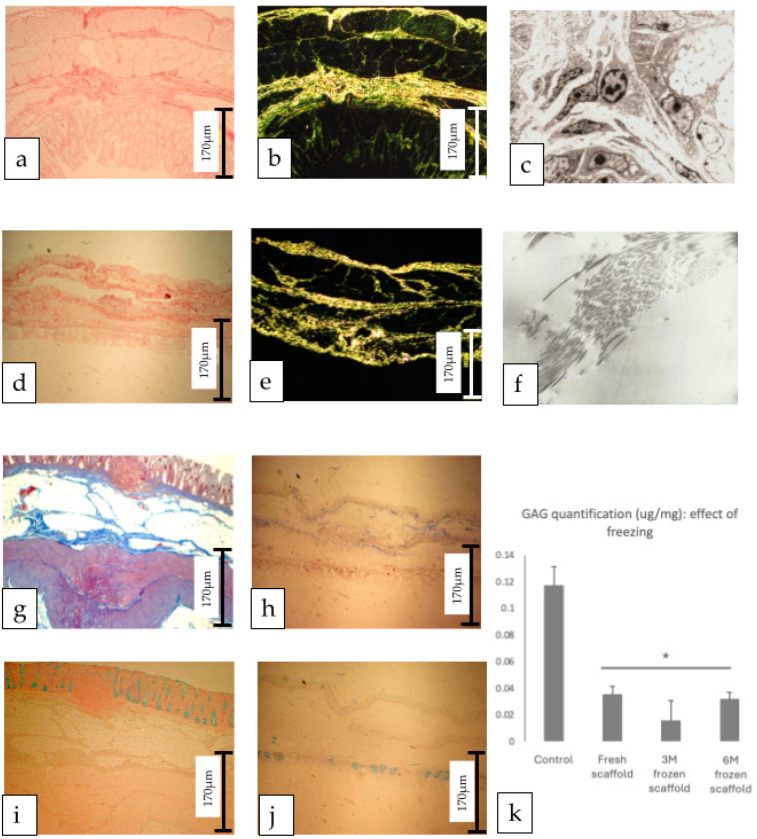
Structural and Biochemical Scaffold Analysis. Control colon stained with Picro-sirius red (PME) demonstrating (**a**) collagen fibres and (**b**) same collagen network viewed under polarised light; (**c**) transmission electron microscopy (TEM) of control tissue showing intact cells and extracellular matrix (ECM); (**d**,**e**) PME staining of scaffold demonstrating preserved ECM (**d**) confirmed by polarised light imaging (**e**); (**f**) higher magnification demonstrating preserved collagen fibres in scaffold; (**g**,**h**) Masson’s Trichrome stain comparing (**g**) control to (**h**) scaffold, indicating retention of elastin fibres; (**i**,**j**) Alcian Blue staining of glycosaminoglycans (GAGs) in (**i**) control colon and (**j**) scaffold matrix, illustrating retention of those molecules in post-decellularisation; (**k**) analysis of the effects of freezing on decellularisation efficiency and GAG content, demonstrating variable reduction (decellularised scaffolds vs control), but no significant differences among decellularised scaffolds, whether fresh or frozen for 3 or 6 months.* statistically significant reduction compared to the control (*p* < 0.05).

**Figure 4 cells-14-00817-f004:**
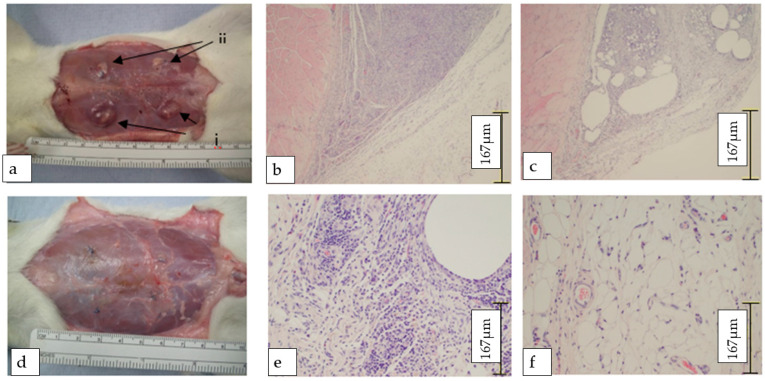
Biocompatibility and In Vivo Response to Scaffold Implants. (**a**) Two-week abdominal implants showing inflammatory response around control colon (‘i’) compared to reduced inflammation around scaffold (‘ii’). (**b**,**c**) Haematoxylin and Eosin staining at two weeks, showing (‘iii’, **b**) dense inflammatory infiltrate in control tissue compared to (**c**) less infiltrated scaffold section. (**d**) Macroscopic view of implants in rats at four weeks. (**e**,**f**) Micrographs showing locale of implants in 20× magnification. Histological comparison at four weeks showing (**e**) intense inflammatory infiltrate in control tissue, contrasting with (**f**) reduced inflammation and evidence of neovascularisation in scaffold implant.

**Figure 5 cells-14-00817-f005:**
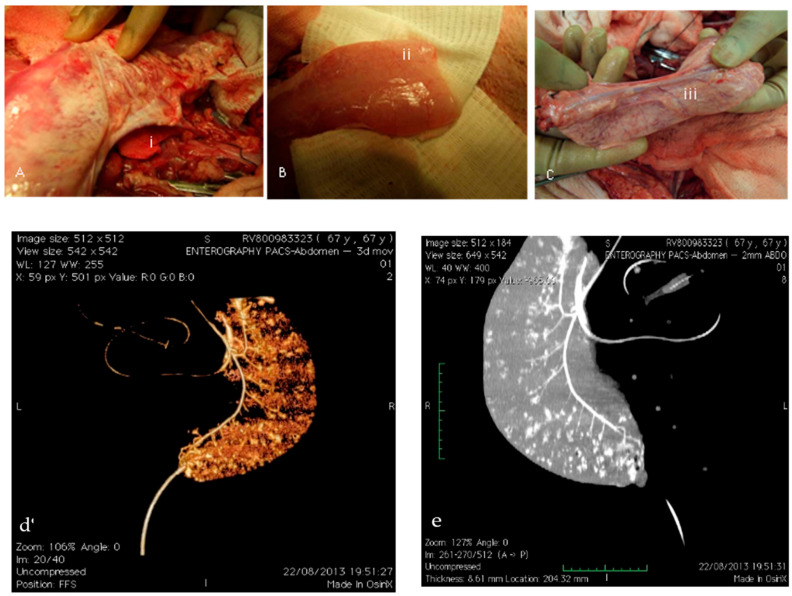
CT Angiographic Evaluation of Vascularised Scaffold. (**a**) Immediate perfusion post-vascular transplantation demonstrated by end-to-end anastomosis with renal artery (‘i’), showing (**b**) evident perfusion of small vessels (‘ii’) and (**c**) scaffold perfusion limited by thrombosis, noted as thrombotic venous occlusion (‘iii’). (**d**) CT angiography reconstruction showing complete small vessel perfusion within scaffold. (**e**) Contrast-enhanced CT angiography identifying small vessel perfusion with minor leakage points within scaffold matrix.

**Table 1 cells-14-00817-t001:** Decellularisation Protocol for Large Intestine—Single Cycle.

Order	Agent (Concentration)	Duration	RPM	Solution Volume
1	Trypsin (0.05%) and EDTA (0.05%)	30 min	30	1 L
2	Distilled Water Wash	15 min	30	1 L
(Wash repeated three times)				
3	DNase I (2KU)	30 min	30	1 L
4	Distilled Water Wash	15 min	30	1 L
(Wash repeated three times)				
5	SDS (0.05%)	30 min	30	1 L
6	Distilled Water Wash	15 min	30	1 L
(Wash repeated three times)				

Decellularisation Protocol Single Cycle: Full decellularisation was achieved by repeating above cycle 5 times. Note considerable use of distilled water to both promote decellularisation and prevent accumulation of cellular debris.

## Data Availability

The original contributions presented in this study are included in the article. Further inquiries can be directed to the corresponding author(s).

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
