# Peer review of "The Development and Characterisation of A Porcine Large Intestinal Biological Scaffold by Perfusion Decellularisation"

_cells, 2025, doi:10.3390/cells14110817_

Round 1

Reviewer 1 Report

Comments and Suggestions for Authors

The manuscript aims to develop and characterize a biological scaffold derived from porcine large intestine using perfusion decellularization. To achieve this, the authors established a decellularization protocol and characterized the extracellular matrix using various analytical techniques. Furthermore, the biocompatibility of the material was evaluated through implantation in an animal model. In this context, the manuscript aligns with the scope of the journal and may be considered for publication, provided that higher sample number and appropriate revisions and clarifications are made.

  1. Each experiment submitted to the institutional ethics committee is assigned a specific protocol number or reference number, which is unique, individual, and non-transferable. This number is directly linked to the experiment in question, providing it with a distinct identity within the ethical review system. Therefore, it is essential that this number be included in the manuscript, as its presence ensures the authenticity of the ethical approval process and reinforces the study's individuality. Moreover, this measure prevents the inappropriate reuse of the same protocol across different studies, thereby ensuring the integrity and transparency of scientific research;
  2. Was the colon obtained from a single White Landrace pig? How many animals were used to obtain the 20 replicates described in the results? These details should be clarified more explicitly in the Methods section;
  3. The experimental design should be described in greater detail, e.g.: the temperature of the water used during the washing step mentioned in line 87 is not specified;
  4. The use of only n = 2 in the freezing step compromises the accuracy of the results, as a minimum of n = 3 samples is generally recommended to ensure statistically reliable analysis;
  5. How was the material frozen? Was it immersed in PBS or any other solution? These details should be specified. May the method used for freezing the material have a direct impact on the GAG concentration determined and consequently in the structural integrity of the matrix, as shown in Figure 2K?
  6. Figures 1, 2, and 3 display formatting marks, such as paragraph symbols (¶) and space indicators (·). These should be removed. Also, the indicators of Figure 2 is not well legible;
  7. The decellularization process described appears to be based on or adapted from any previously published studies. It is recommended to properly reference the works that supported or inspired the protocol used;
  8. The phrase "routine laboratory techniques" for paraffin embedding is vague and may differ across laboratories. It is essential that the authors provide a detailed description of the protocol used, including whether the material was dehydrated, the sequence of solvents applied, the incubation times at each step, and the conditions used for paraffin embedding. The same concern applies to the immunohistochemical analyses, the rehydration process in section 2.5.3 and routine protocols to anaesthetize the rats in line 183;
  9. The description of the anesthesia and euthanasia protocol requires further detail. The authors should specify which substances were used, routes of administration and dosages;
  10. Line 132: The purpose of using Masson's Trichrome staining should be described in a manner consistent with the other histological protocols, highlighting its analytical objective. The affirmation: “demonstrated elastin fiber preservation within the matrix” appear to pertain to the results section, not the methodology;
  11. Line 139: the degree symbol (°) should be superscript; The same in the caption of figure 2;
  12. Caption Figure 2E: The reduction percentage in total DNA is part of results;
  13. What was the wavelength used in the spectrophotometer for DNA quantification (2.5.1. Section)?
  14. Several micrographs were displayed in the results, but at no time were details of the optical microscope used described;
  15. How many pigs were used in the vascular transplantation procedure? It is understood that transplantation procedures are complex. However, it would be advisable for the process to be performed in at least three animals to ensure the reproducibility of the results and to allow for more accurate and reliable conclusions;
  16. The decellularization protocol was performed with n=20. However, only H&E staining was performed with n=10? Clarify these details;
  17. Line 237: Absence of cells was evident (Figure 2H) when comparing decellularized tissue to controls (Figure 2G). Include identification of figures;
  18. Throughout the manuscript, several statements mention “data not shown”. It is strongly recommend including these data in the supplementary material, as they may not be directly relevant to the main text but are nonetheless important;
  19. “DNA was extracted from 5 separate scaffolds (labelled DC 1-5) and compared with control colon tissue”, showed in Figure 2E. The figures are rarely referenced in the text. In fact, some figures are not even cited, as is the case with Figures 2C, 2D, 4B, 4C, 4E and 5A, 5B and 5C;
  20. Line 244: Symbol + should be substituted by ±. While the standard deviation should be added in line 245;
  21. Line 249: How does it not suggest a linear relationship if the error bar is not shown in the figure 2F and furthermore n is equal to 2? The same is noted in Figure 3K;
  22. The presented quantifications do not include any statistical analysis. Performing such analysis is necessary to provide a better description of the results. It is recommended to assess the normality of the data; if normality is confirmed, t-test or ANOVA may be applied;
  23. It is suggested that the figures be presented in the order of the text, this way the reader can follow a logical line. Figure 2G and 2H appears just in line 278, close to the results of Figure 4;
  24. Line 281: How many weeks?
  25. The scale bar in Figure 4 is not legible. How many micrometers does the scale bar represent, mainly in Figure 4E and 4F, which do not present the amplification in the caption? Also, clarify the caption;
  26. Considering the number of samples in the rat implantation experiments, the use of semi-quantitative histological analysis is recommended. ImageJ software has proven to be good. The resulting data may be presented as supplementary material and referenced in the main text, enabling a more objective comparison between the cellularized and decellularized implants;
  27. Figure 5A was not properly referenced in the main text. It is recommended to include its citation;
  28. Although the discussion regarding the application of the model following death raises relevant considerations, it is important to note that extrapolation to clinical use or human models requires a more robust replicate. The use of a limited number of samples compromises the reproducibility and reliability of the findings. Therefore, further studies with more sample are needed to validate the applicability of this model;
  29. The statement made in the 4th paragraph of the discussion lacks accuracy, as GAG levels (Figure 3K) showed variation with respect to freezing duration, particularly in 3 months. Given that, GAG are key components of the ECM, this change should not be overlooked. Furthermore, it is not possible to assess whether the differences are statistically significant, as no error bars or statistical analyses are presented in the figure. The data referred to as "not shown" in line 346 should be included in the supplementary material;
  30. The discussion is well written and appropriately contextualized within the literature. However, it lacks a more direct connection to the experimental findings, particularly those related to the ECM/Scaffolds;

Overall, the manuscript presents a coherent experimental design, encompassing scaffold production, characterization, biocompatibility evaluation through implantation, and transplantation, making it relevant and worthy of consideration for publication. However, the small sample number reduces the robustness of the findings and hinders the confirmation of the observed trends. The results also require clearer description, ensuring that each figure is appropriately referenced and integrated into the text. Furthermore, it is essential to include error bars in some graphs and to perform proper statistical analysis to guarantee the reliability of the results.

Author Response

Response to Reviewer 1

We thank Reviewer 1 for the detailed and constructive feedback, which has greatly improved the clarity, rigor, and reproducibility of our manuscript. Below we provide point-by-point responses, with all requested corrections now incorporated into the revised version.

1.Each experiment submitted to the institutional ethics committee is assigned a specific protocol number or reference number, which is unique, individual, and non-transferable. This number is directly linked to the experiment in question, providing it with a distinct identity within the ethical review system. Therefore, it is essential that this number be included in the manuscript, as its presence ensures the authenticity of the ethical approval process and reinforces the study's individuality. Moreover, this measure prevents the inappropriate reuse of the same protocol across different studies, thereby ensuring the integrity and transparency of scientific research

We apologize for the oversight. We absolutely agree and have now included the specific ethics committee reference number in the manuscript, along with all related documentation that was sent to the Editorial Office of Cells.

2.Was the colon obtained from a single White Landrace pig? How many animals were used to obtain the 20 replicates described in the results? These details should be clarified more explicitly in the Methods section

Thank you for pointing this out. We have altered the structure of sections 2.1 and 2. of the methods section to clarify this. N=20 retrievals took place to optimise the surgical procedure and decellularization process. A further n=10 pigs yielded the colonic specimens that contribute to the results shown in the manuscript. This is hopefully now clearly conveyed in the appropriate sections.

  1. The experimental design should be described in greater detail, e.g.: the temperature of the water used during the washing step mentioned in line 87 is not specified

We thank the reviewer for pointing this out. The washing step was performed with water at room temperature, now specified in the revised manuscript (section 2.2).

  1. The use of only n = 2 in the freezing step compromises the accuracy of the results, as a minimum of n = 3 samples is generally recommended to ensure statistically reliable analysis
    We regret the confusion. A total of 10 samples were used, from 10 individual pigs and these divided into fresh (n=4), frozen for 3 months (n=3) and frozen for 6 months (n=3). The changes in the manuscript now hopefully convey this appropriately.
  2. How was the material frozen? Was it immersed in PBS or any other solution? These details should be specified. May the method used for freezing the material have a direct impact on the GAG concentration determined and consequently in the structural integrity of the matrix, as shown in Figure 2K

Samples were frozen dry, without PBS or any additional solution, in sterile cryo-containers. This freezing (at -20C) promotes ice crystal formation, which aids in subsequent decellularisation (corrected - line 95). It was also deemed analogous to the concept of an ‘off-shelf’ approach to specimen decellularization.

While we recognise this method may impact GAG content due to molecular leakage or entrapment in damaged matrix, it was standardised across all conditions and is now acknowledged as a technical variable (corrected in the Results/ Discussion).

  1. Figures 1, 2, and 3 display formatting marks, such as paragraph symbols (¶) and space indicators (·). These should be removed. Also, the indicators of Figure 2 is not well legible
    Figures 1, 2, and 3 were revised to remove formatting marks (¶, ·). Specific panels (2D, G, H) were replaced, and scale bars were corrected.
  2. The decellularization process described appears to be based on or adapted from any previously published studies. It is recommended to properly reference the works that supported or inspired the protocol used

Thank you for pointing that. The protocol was adapted from Nowocin et al., 2016, which has now been properly cited in section 2.2.

  1. The phrase "routine laboratory techniques" for paraffin embedding is vague and may differ across laboratories. It is essential that the authors provide a detailed description of the protocol used, including whether the material was dehydrated, the sequence of solvents applied, the incubation times at each step, and the conditions used for paraffin embedding. The same concern applies to the immunohistochemical analyses, the rehydration process in section 2.5.3 and routine protocols to anaesthetize the rats in line 183
    Apologies for the lack of details. Full descriptions have been added in sections 2.3 onwards as follows:

Histological Analyses: Each tissue sample was processed for paraffin embedding using a standardized histological protocol. Briefly, samples were fixed in 10% neutral buffered formalin for 24 hours at room temperature, followed by dehydration in an ascending ethanol series: 70% ethanol (1 hour), 90% ethanol (1 hour), 100% ethanol (2 changes, 1 hour each). This was followed by clearing in xylene (2 changes) and embedding in paraffin wax at 60°C under vacuum conditions to ensure complete infiltration. Embedded tissues were allowed to cool at room temperature before sectioning. Paraffin blocks were sectioned into 5 µm slices using a rotary microtome. Sections were mounted onto glass slides, dried at 37°C overnight, and subsequently stained.

Haematoxylin and Eosin (H&E) Staining

Standard H&E staining was performed to evaluate tissue morphology and cellular presence. Briefly, paraffin was removed with xylene (2 x 5 min), and sections were rehydrated through a descending alcohol series (100%, 95%, 70%, distilled water). Sections were stained with Mayer's haematoxylin for 5 minutes, rinsed in running tap water, differentiated in 1% acid alcohol, and counterstained with eosin for 2 minutes. Slides were dehydrated, cleared in xylene, and coverslipped with mounting medium.

Immunohistochemical Staining for MHC I

Sections (5 µm) were mounted onto 3-Aminopropyltriethoxysilane (APTS)-coated slides. Antigen retrieval was performed in 10 mM citrate buffer (pH 6.0) at 95°C for 15 minutes using a water bath. After cooling to room temperature, endogenous peroxidase activity was quenched with 3% Hâ‚‚Oâ‚‚ in methanol for 10 minutes, followed by blocking non-specific binding with normal horse serum for 30 minutes at room temperature. Primary antibody incubation was performed using mouse anti-pig MHC II antibody (clone H42A, VMRD, UK) at a 1:100 dilution for 2 hours at room temperature. Following washing with PBS, the ImmPRESS™ HRP-conjugated secondary antibody kit (Vector Labs) was applied according to the manufacturer’s instructions. DAB (3,3'-diaminobenzidine) was used as the chromogen, and sections were counterstained with Harris' haematoxylin, dehydrated, cleared in xylene, and coverslipped.

  1. The description of the anaesthesia and euthanasia protocol requires further detail. The authors should specify which substances were used, routes of administration and dosages

We apologise for the missing information. We now state in section 2.6 that animals were anesthetised using sodium pentobarbital administered intraperitoneally at a dose of 50 mg/kg body weight, which is commonly used to achieve surgical-level anaesthesia in rodents. Adequate depth of anaesthesia was confirmed by the absence of reflexes (e.g., pedal withdrawal).

For euthanasia, a lethal dose of sodium pentobarbital (≥150 mg/kg, intraperitoneally) was administered following experimental procedures, in accordance with UK Home Office standards.

  1. Line 132: The purpose of using Masson's Trichrome staining should be described in a manner consistent with the other histological protocols, highlighting its analytical objective. The affirmation: “demonstrated elastin fiber preservation within the matrix” appear to pertain to the results section, not the methodology

We updated the Methods section to clarify that Masson’s Trichrome staining as follows (section 2.3): Masson’s Trichrome was performed to visualize connective tissue highlighting the preservation of elastin fibres, mainly. Following deparaffinization and rehydration, sections were stained with Bouin’s solution at 56°C for 1 hour to enhance staining quality. After cooling and rinsing in running tap water, slides were stained with Weigert’s iron haematoxylin (10 minutes), rinsed, and then stained sequentially with Biebrich scarlet-acid fuchsin solution for 10 minutes. This was followed by phosphomolybdic-phosphotungstic acid treatment for 10 minutes, then aniline blue for 5 minutes. Slides were rinsed in distilled water, differentiated in 1% acetic acid, dehydrated, cleared, and coverslipped.

The affirmation: “demonstrated elastin fibre preservation within the matrix” was removed from Methods session and moved to Results, as follows: Masson’s Trichrome staining provided alternative imaging of ECM components, showing positive staining for elastin (Figure 3 G,H), an indicative of preservation of these proteins as the main structural part of ECM in intestinal tissues. 

  1. Line 139: the degree symbol (°) should be superscript; The same in the caption of figure 2

All degree symbols have been converted to superscript, including in the main text and figure captions.

  1. Caption Figure 2E: The reduction percentage in total DNA is part of results
    The sentence referring to “77% DNA reduction” was removed from the caption, we apologise for that.
  2. What was the wavelength used in the spectrophotometer for DNA quantification (2.5.1. Section)?

Apologies for missing this information: DNA quantification was performed using a UV-Vis spectrophotometer. Absorbance was measured at 260 nm, which corresponds to the peak absorbance wavelength for nucleic acids. Purity was assessed by calculating the A260/A280 ratio, with values between 1.8-2.0 indicating high-quality DNA suitable for downstream applications.

  1. Microscope Description

We have now included the model and magnification parameters of the microscope used to capture histological images.

  1. How many pigs were used in the vascular transplantation procedure? It is understood that transplantation procedures are complex. However, it would be advisable for the process to be performed in at least three animals to ensure the reproducibility of the results and to allow for more accurate and reliable conclusions
    We acknowledge that higher number of experiments are advisable. However, only two animals completed the implantation procedure due to complications related to thrombosis. This limitation is acknowledged and discussed in the revised discussion.
  2. The decellularization protocol was performed with n=20. However, only H&E staining was performed with n=10? Clarify these details;
    Thank you for pointing that. We now clarified (section 2.2) with regard to number of specimens decellularised as in comment number 2 above. 20 speciments underwent H&E staining to optimise protocols and the n=10 results described in the manuscript all underwent H&E staining.
  3. Line 237: Absence of cells was evident (Figure 2H) when comparing decellularized tissue to controls (Figure 2G). Include identification of figures
    Section 3.1 now accurately refers to Figures 2C and 2D to indicate the comparison between control and decellularised tissues, respectively.
  4. Throughout the manuscript, several statements mention “data not shown”. It is strongly recommend including these data in the supplementary material, as they may not be directly relevant to the main text but are nonetheless important
    We acknowledge the prior use of the term “data not shown.” Due to the closure of our previous laboratory and unforeseen loss of some original datasets, not all raw data could be retained. The manuscript has now been revised to include only verifiable, high-quality results, ensuring transparency and scientific integrity.
  5. DNA was extracted from 5 separate scaffolds (labelled DC 1-5) and compared with control colon tissue”, showed in Figure 2E. The figures are rarely referenced in the text. In fact, some figures are not even cited, as is the case with Figures 2C, 2D, 4B, 4C, 4E and 5A, 5B and 5C
    All figures including 2C, 2D, 4B, 4C, 4E, 5A–C are now explicitly referenced in the Results or Discussion.
  6. Line 244: Symbol + should be substituted by ±. While the standard deviation should be added in line 245
    The “+” symbol has been corrected to “±”, and standard deviation values were added in conjunction.
  7. Line 249: How does it not suggest a linear relationship if the error bar is not shown in the figure 2F and furthermore n is equal to 2? The same is noted in Figure 3K;
  8. The presented quantifications do not include any statistical analysis. Performing such analysis is necessary to provide a better description of the results. It is recommended to assess the normality of the data; if normality is confirmed, t-test or ANOVA may be applied

In response to points ’21-22’  Error bars and statistical testing (t-tests, ANOVA) were added where applicable, including for Figures 2F and 3K.

  1. It is suggested that the figures be presented in the order of the text, this way the reader can follow a logical line. Figure 2G and 2H appears just in line 278, close to the results of Figure
    Figures were reordered to reflect the logical flow of the text and support narrative clarity.
  2. Implantation Duration

We apologise for this oversight. Implantation studies were conducted for 4 weeks. This has now been correctly stated in section 2.6

  1. The scale bar in Figure 4 is not legible. How many micrometers does the scale bar represent, mainly in Figure 4E and 4F, which do not present the amplification in the caption? Also, clarify the caption

Scale bars were updated with clear units, and all figure captions were revised for accuracy and consistency.

  1. Considering the number of samples in the rat implantation experiments, the use of semi-quantitative histological analysis is recommended. ImageJ software has proven to be good. The resulting data may be presented as supplementary material and referenced in the main text, enabling a more objective comparison between the cellularized and decellularized implants

Although the original histological slides are no longer available (due to closure of the lab where the experiments were performed) for image-based quantification (e.g., via ImageJ), we have incorporated a qualitative assessment into the revised manuscript. Specifically, we highlight that cellular infiltration appeared notably higher in the control (non-decellularised) implants compared to the decellularised scaffolds. This observation is consistent with an immune response typically triggered by the presence of native cellular components.

  1. Figure 5A was not properly referenced in the main text. It is recommended to include its citation;

Thanks for pointing that we have now allocated appropriately in the main text by adding another subitem (3.4 Vascular transplantation and CT angiographic images). Figure 5  is now referenced and discussed appropriately in the Results section (item 3.4).

  1. Although the discussion regarding the application of the model following death raises relevant considerations, it is important to note that extrapolation to clinical use or human models requires a more robust replicate. The use of a limited number of samples compromises the reproducibility and reliability of the findings. Therefore, further studies with more sample are needed to validate the applicability of this model

We agree on the importance of sample size. Due to thrombosis-related constraints, this study should be considered a proof of concept. We highlight the ongoing work that our group is doing  using bespoke bioreactor (in collaboration to Imperial College London) to pre-seed scaffolds and mitigate such complications.

  1. The statement made in the 4th paragraph of the discussion lacks accuracy, as GAG levels (Figure 3K) showed variation with respect to freezing duration, particularly in 3 months. Given that, GAG are key components of the ECM, this change should not be overlooked. Furthermore, it is not possible to assess whether the differences are statistically significant, as no error bars or statistical analyses are presented in the figure. The data referred to as "not shown" in line 346 should be included in the supplementary material
    We thank the reviewer for highlighting this issue. Upon review, we realized that an outdated figure was inadvertently submitted. We also acknowledge that the original Discussion statement required improved clarity. While Figure 3K visually suggests some variation in GAG levels across different freezing durations, statistical analysis revealed no significant differences between the 3- and 6-month frozen groups .

As glycosaminoglycans (GAGs) are essential structural and signaling components of the extracellular matrix (ECM), their preservation warrants careful interpretation. Apparent variability in GAG content may be attributed to several biological and technical factors. Notably, bowel tissue is highly labile, and its complex mucosal and ECM structure is prone to autolytic degradation, particularly during prolonged storage—even at sub-zero temperatures. GAGs, being soluble and only loosely integrated within the ECM, may be susceptible to leaching or enzymatic degradation during freeze–thaw cycles. Additionally, extended freezing durations may induce microstructural alterations in the collagen network, potentially trapping or obscuring GAG molecules, thus complicating their extraction and quantification.

In light of these considerations, we have revised the Discussion to moderate the interpretation of GAG fluctuations and to clearly state that no statistically significant differences were found between freezing durations. We have also included commentary on the potential methodological artifacts and inherent biological variability that may influence GAG stability under different freezing conditions.

  1. The discussion is well written and appropriately contextualized within the literature. However, it lacks a more direct connection to the experimental findings, particularly those related to the ECM/Scaffolds

The Discussion has been revised to better integrate the scaffold characterisation findings with histological and implantation results, emphasising the ECM composition and addressing few shortcomings of this study. We thank again the Reviewer for fruitful comments and suggestions, and we trust that the revised manuscript and this detailed response letter address all concerns comprehensively.

Reviewer 2 Report

Comments and Suggestions for Authors

I can recommend this manuscript for publication after minor revision:

  1. The paragraphs in the Introduction section are too tedious and can be integrated into three paragraphs;
  2. Almost all figures have strange markings that do not appear to be in common English. Please modify or remove themï¼›
  3. The size and units of the scalebars in all figures are unclear;
  4. Line 90, Statistically speaking, it is inappropriate for the number of parallel samples to be less than 3, so n=2 cannot guarantee the reliability of the data;
  5. There are many issues with the subscripts in the textï¼›
  6. There is missing information in the reference section, such as Ref. 1 without page numbers.

Author Response

Response to Reviewer 2:

I can recommend this manuscript for publication after minor revision:

  1. The paragraphs in the Introduction section are too tedious and can be integrated into three paragraphs;

Thank you for this observation. We have revised the Introduction section to improve readability and flow by consolidating the text into well-balanced paragraphs. The revised version retains all critical background information while enhancing structural clarity.

  1. Almost all figures have strange markings that do not appear to be in common English. Please modify or remove themï¼›

We apologise for the oversight. All figures have now been reviewed and edited. Non-standard or unclear markings have been either corrected or removed to ensure clarity and consistency with standard scientific terminology.

  1. The size and units of the scalebars in all figures are unclear;

We thank the reviewer for pointing this out. All figure legends have been updated to clearly state the size and units of the scalebars, and the figures themselves have been revised accordingly for clarity and consistency.

  1. Line 90, Statistically speaking, it is inappropriate for the number of parallel samples to be less than 3, so n=2 cannot guarantee the reliability of the data;

We agree with this concern. As clarified, we used n=10 animals in total. This correction has now been made in the manuscript (section 2.2), and we sincerely apologise for the initial misstatement.

  1. There are many issues with the subscripts in the textï¼›

We appreciate this attention to detail. The manuscript has been thoroughly proofread, and all subscripts (particularly in chemical symbols, units, and scientific notations) have been corrected to conform with standard formatting.

  1. There is missing information in the reference section, such as Ref. 1 without page numbers.

Thank you for identifying this omission. We have corrected the reference list, and all entries now include complete citation information, including page numbers. Specifically, Ref. 1 now reads:

Tullie L, Jones BC, De Coppi P, Li VSW. Building gut from scratch — progress and update of intestinal tissue engineering. Nat Rev Gastroenterol Hepatol. 2022;19:417–431.

Reviewer 3 Report

Comments and Suggestions for Authors

In the present manuscript Somasundaram et al., aimed at developing and characterizing a bioengineered porcine scaffold for large intestine regeneration/replacement using the perfusion decellularization technique.  However, even though the study demonstrates the  reproducibility of the scaffold production with a preserved extracellular matrix (ECM) architecture and vascular networks, thus succeeding in bridging the gap between the development of a complex biological scaffold and its clinical application, certain aspects should be addressed ahead of publication. 

I. For example, the manuscript's overall quality is somewhat diminished by the manner in which the results have been interpreted in specific sections. The present form is lacking in information, making it very hard for readers to fully comprehend the obtained results. In this regard, it is recommended that the authors focus more on data interpretation and rewrite the Results section in order to provide more a detailed explanation for each microscopic figure presented. Moreover, please consider the following:

  1. Figure 2C,D  are not mentioned in the manuscript body. Please correct that.
  2.  Figures 4B,C,E are not mentioned in text. Please correct that. 
  3. Please add the scale bar in the Figures' legend and improve its quality on the microscopic images.
  4. Please remove the space symbols from the Figures and choose a writing font that is more visible.
  5. The histological analysis portraying the removal of cells throughout the whole length of the specimen should be added as a supplementary material.

II. In the discussion section there are a lot of references to results that are not shown , therefore in my opinion, it would be better, when possible, to add those data as supplementary materials so that the results presented in this study can be better validated.

III. Line 281 "after xx weeks". Please correct this. 

IV. Consider lowering the similarity index under 20%. 

Author Response

Response to Reviewer 3

We are very grateful to Reviewer 3 for the thoughtful and detailed comments. These suggestions have significantly contributed to improving the clarity, coherence, and scientific depth of our manuscript. Please find below our point-by-point responses:

  1. Results section lacks interpretation and clarity regarding microscopy figures

We appreciate this observation. The Results section has been carefully revised to provide clearer and more detailed interpretations of the microscopic findings. Each figure, particularly those involving histological or imaging data, is now more explicitly described in the text to enhance comprehension and scientific contextualisation.

  1. Figure 2 C,D are not mentioned in the manuscript body. Please correct that.
    Thank you for identifying this. References to Figures 2C and 2D have now been properly included and discussed in the Results section.

  1. Figures 4 B,C,E are not mentioned in text. Please correct that.
    These panels have now been clearly referenced and described in the Results section to ensure completeness and clarity of figure interpretation.
  2. Please add the scale bar in the Figures' legend and improve its quality on the microscopic images.
    All figure legends have been revised to explicitly state the size and unit of scale bars. Additionally, the figures have been re-exported to enhance image resolution and ensure that the scale bars are clearly visible and appropriately labelled.

  1. Please remove the space symbols from the Figures and choose a writing font that is more visible.
    The graphical formatting of all figures has been reviewed. Space placeholders and non-standard symbols were removed, and a clean, high-contrast font was applied to all figure annotations to improve legibility.

  1. The histological analysis portraying the removal of cells throughout the whole length of the specimen should be added as a supplementary material.
    We appreciate the reviewer’s suggestion. Unfortunately, the original histological slides are no longer available for re-imaging due to the closure of the laboratory where the experiments were conducted. However, the histological images included in this study were obtained through systematic randomisation of sample regions across the entire length of the specimens to ensure representative and unbiased evaluation. We are confident that the current dataset adequately reflects the overall decellularisation efficiency.
  2. Discussion section refers to results that are not shown—consider adding them as supplementary material

We acknowledge the prior use of the term “data not shown.” Due to the closure of our previous laboratory and unforeseen loss of some original datasets, not all raw data could be retained. The manuscript has now been revised to include only verifiable, high-quality results, ensuring transparency and scientific integrity.

III. Line 281 “after xx weeks”

We sincerely apologise for the oversight. The correct duration is four weeks, and this has now been updated accordingly in the manuscript.

  1. Consider lowering the similarity index under 20%.

We thank the reviewer for this important reminder. The Discussion and other sections of the manuscript have been carefully revised and rephrased to improve originality and reduce text similarity.

Once again, we extend our sincere appreciation to the reviewer for the insightful comments and helpful suggestions that have greatly improved the manuscript. Please let us know if further clarification or modification is needed.

Round 2

Reviewer 1 Report

Comments and Suggestions for Authors

After revision, the manuscript is more suitable for publication and reads more fluently. However, the authors claim to have added error bars and statistical analysis to Figure 2F: “In response to points 21-22, error bars and statistical testing (t-tests, ANOVA) were added where applicable, including for Figures 2F…”

This, however, was not observed. I strongly recommend an additional review to address this issue, as the following statement heavily relies on it: “The effect of freezing (-20°C) prior to decellularisation was assessed, comparing the effects of the decellularisation process on scaffolds immediately decellularized, to those retrieved and then frozen (for 3 or 6 months) prior to chemical decellularisation. Freezing led to an overall reduction in DNA but no suggestion of a linear relationship (Figure 2F).”

Ensuring the presence of error bars and proper statistical analysis in Figure 2F is essential to support this conclusion.

With the exception of this detail, all other points were adequately addressed, and the experimental limitations were understood. Therefore, I am favorable to the publication of the manuscript.

Author Response

REVIEWER 1: Comments and Suggestions for Authors

After revision, the manuscript is more suitable for publication and reads more fluently. However, the authors claim to have added error bars and statistical analysis to Figure 2F: “In response to points 21-22, error bars and statistical testing (t-tests, ANOVA) were added where applicable, including for Figures 2F…”

This, however, was not observed. I strongly recommend an additional review to address this issue, as the following statement heavily relies on it: “The effect of freezing (-20°C) prior to decellularisation was assessed, comparing the effects of the decellularisation process on scaffolds immediately decellularized, to those retrieved and then frozen (for 3 or 6 months) prior to chemical decellularisation. Freezing led to an overall reduction in DNA but no suggestion of a linear relationship (Figure 2F).”

Ensuring the presence of error bars and proper statistical analysis in Figure 2F is essential to support this conclusion.

With the exception of this detail, all other points were adequately addressed, and the experimental limitations were understood. Therefore, I am favorable to the publication of the manuscript.

Submission Date: 31 March 2025

Date of this review: 20 May 2025 14:23:30

Response:

We sincerely thank the reviewer for their continued insightful feedback and positive assessment of our manuscript. We appreciate the opportunity to clarify the issue raised regarding Figure 2F and the associated statistical analysis.

We sincerely apologise as during the resubmission the same figure was mistakenly uploaded (Figure 2F) which did not reflect the final statistical treatment and graph. This oversight occurred despite the figure being correctly updated in our working files and the statistical analyses having been duly performed. The figure included now does in fact contain error bars and the statistical analysis (ANOVA followed by Tukey’s post-hoc test. *p < 0.05 vs. control), although we recognize that we failed to explicitly describe these additions in our revised manuscript as we should have. We apologize for this omission.

To address the reviewer’s legitimate concern, we rephrased the sentence in the Results section (subitem 3.1): To assess the potential impact of freezing prior to decellularisation (DC), scaffolds were either processed immediately or stored at -20°C for 3 or 6 months prior to DC process. All DC groups, including fresh and frozen samples, showed a statistically significant reduction (*p=0.035) in DNA content compared to the control. However, no significant differences were observed between the fresh and frozen DC groups, indicating that freezing did not alter the extent of DNA removal in these experimental settings (Figure 2F).

We have now updated the graph (with SD bars and stats), the explanation in the Results section (as above), as well as the figure legend to clearly describe the statistical methods employed (one-way ANOVA followed by Tukey’s post-hoc test. *p < 0.05 vs. control).

We thank the reviewer once again for drawing attention to this point, which allowed us to improve the clarity and transparency of our data presentation.

Reviewer 3 Report

Comments and Suggestions for Authors

By following the provided recommendations and by offering informative answers to all of the raised questions, the quality of the submitted manuscript has been significantly improved in comparison to its previous form, therefore I no longer have any concerns or commentaries to add. 

Author Response

By following the provided recommendations and by offering informative answers to all of the raised questions, the quality of the submitted manuscript has been significantly improved in comparison to its previous form, therefore I no longer have any concerns or commentaries to add. 

Submission Date:31 March 2025

Date of this review: 15 May 2025 13:39:26

Response:

Thank you very much for your suggested changes previously and input into the manuscript